# `RecCrysFormer`: Refined Protein Structural Prediction from 3D Patterson Maps via Recycling Training Runs

Tom Pan[1], Evan Dramko[1], Mitchell D. Miller[2], George N. Phillips, Jr.[2,3], Anastasios Kyrillidis[1,4]

[1]Department of Computer Science [2]Department of BioSciences [3]Department of Chemistry
[4]Ken Kennedy Institute
Rice University

Determining protein structures at an atomic level remains a significant challenge in structural biology. We introduce `RecCrysFormer`, a hybrid model that exploits the strengths of transformers with the aim of integrating experimental and ML approaches to protein structure determination from crystallographic data. `RecCrysFormer` leverages Patterson maps and incorporates known standardized partial structures of amino acid residues to directly predict electron density maps, which are essential for constructing detailed atomic models through crystallographic refinement processes. `RecCrysFormer` benefits from a "recycling" training regimen that iteratively incorporates results from crystallographic refinements and previous training runs as additional inputs in the form of template maps. Using a preliminary dataset of synthetic peptide fragments based on Protein Data Bank, `RecCrysFormer` achieves good accuracy in structural predictions and shows robustness against variations in crystal parameters, such as unit cell dimensions and angles.

## 1. Introduction

**Background.** Proteins are fundamental components of biological processes, acting as molecular machines within our cells [1]. They are polymers composed of small organic molecules called amino acids, linked by peptide bonds. There are 20 standard proteinogenic amino acids, and a single amino acid is referred to as a residue. Amino acid polymers fold to form intricate 3D structures, and understanding these is pivotal as the 3D conformation of a protein largely determines its functionality. Traditional experimental methods for protein structure determination include X-ray crystallography, NMR, and cryo-electron microscopy; see [2]. These methods face a classic inverse problem in science: reconstructing a complete structure from incomplete experimental information.

**The role of machine learning (ML).** Recent years have seen the emergence of ML as another powerful tool in protein structure prediction. Research projects like AlphaFold2 [3] have demonstrated the potential of deep learning in achieving highly accurate predictions by leveraging protein structural data alongside co-evolutionary information (e.g. multiple sequence alignments).

X-ray crystallography remains widely used for its ability to provide accurate atomic coordinates, including interactions with small molecules and metal ions. *With this work, we aim to help bridging the gap between experimental crystallographic methods and ML techniques by developing a prototype to directly translate X-ray diffraction patterns of protein crystals into solved structures.*

**Motivation and contributions.** We introduce `RecCrysFormer` that combines convolutional layers with a 3D vision transformer. This model integrates domain-specific knowledge with established ML architectures to address a fundamental problem in structural biology. Key features include:

- The use of Patterson maps, directly obtainable from experimental data, and "partial structure" densities corresponding to the most common conformation of individual residues.
- The integration with established crystallographic refinement procedures, such as *SHELXE* [4, 5], which are applied to our predicted electron density maps to obtain protein structure coordinates.
- A "recycling" meta-algorithm that enhances training by reusing the outputs from previous training (potentially post-refinement) as template features in subsequent iterations.

Second Conference on Parsimony and Learning (CPAL 2025).

- The development of a dataset comprised of synthetic peptide fragments based on PDB entries with varied unit cell sizes and angles. Our prototypical model shows robustness against structural variations and delivers good post-training predictions.

While this work focuses on small synthetic protein fragments that also have higher solvent content, it represents a step toward applying ML to complex, real-world crystallographic problems.

## 2. Problem background and related works

**X-ray crystallography and the phase problem.** X-ray crystallography, a century-old technique, prominently serves in determining protein structures through mapping the electron density within crystals [6]. This method involves irradiating protein crystals with X-rays, which diffract based on a crystal's internal structure. Each repeating unit (unit cell) in the crystal typically contains identical molecular arrangements; this causes the X-rays to scatter constructively or destructively and form output beams only in certain directions. These diffracted beams then produce a pattern of spots, called reflections, on a detector. A reflection is characterized by its Miller indices $(h, k, l)$, which indicate the orientation of planes within the unit cell contributing to producing the reflection [7].

The mathematical representation of a reflection, known as the structure factor $F(h, k, l)$, encapsulates the sum of atomic contributions within the unit cell:

$$F(h, k, l) = \sum_{j=1}^{n} f_j \cdot e^{2\pi i (h x_j + k y_j + l z_j)}, \tag{1}$$

where $f_j$ denotes the scattering factor and $(x_j, y_j, z_j)$ the coordinates of the $j$-th atom. A structure factor comprises an amplitude and a phase $\phi(h, k, l)$, both needed for reconstructing the electron density $\rho(x, y, z)$ at all locations within the crystal's unit cell via a Fourier transform:

$$\rho(x, y, z) = \frac{1}{V} \sum_{h,k,l} |F(h, k, l)| \cdot e^{-2\pi i (hx + ky + lz - \phi(h,k,l))}, \tag{2}$$

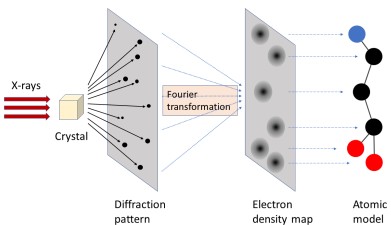

where $V$ is the volume of the unit cell. Although the amplitude $|F(h, k, l)|$ of a structure factor is directly measurable, from the intensity of the corresponding diffraction spot, the phase $\phi(h, k, l)$ is not. This is the crystallographic phase problem, a fundamental challenge for directly using X-ray crystallography data for accurate electron density mapping [6]. See also Figure 1.

Figure 1: A representation of the process for determining crystal structures. Complete structure factors are obtained from diffraction patterns through various methods. By applying a Fourier transform to these, the electron density within the unit cell is calculated. The initial model is then iteratively refined through comparison with experimental measurements.

**Patterson maps.** Critical to diffraction interpretation is the Fourier transform from full structure factors to electron density. A key intermediary in several methods for obtaining these is the *Patterson function*, which modifies the previous Fourier transform by squaring amplitudes and nullifying phase information, creating a Patterson map [8]:

$$p(u, v, w) = \frac{1}{V} \sum_{h,k,l} |F(h, k, l)|^2 \cdot e^{-2\pi i (hu + kv + lw)}. \tag{3}$$

Here, $(u, v, w)$ represents coordinates within the Patterson map's unit cell, mirroring the exact dimensions of the crystal's unit cell. The Patterson function's design inherently omits phase data, allowing computation directly from raw diffraction measurements. Yet, a Patterson map does not directly illustrate atomic locations with a unit cell, but rather indicates vectorial relationships between atoms. Each peak in a Patterson map corresponds to an interatomic vector between atoms within the crystal's unit cell, and so the number of peaks scales quadratically with the original amount of atoms. Additionally, the presence of heavy atoms can dominate the map, as the height of peaks is proportional to the product of atomic numbers in the corresponding pair. Thus, they are not directly used to estimate electron densities.

**Related works.** The most widely used approaches to obtain structure factors include isomorphous replacement, anomalous scattering, and molecular replacement. Each presents unique challenges and requirements [6, 9]. While isomorphous replacement and anomalous scattering necessitate multiple experimental conditions and the incorporation of heavy atoms into the desired structure, molecular replacement relies on the availability of closely related homologous structures or accurately predicted models, which are not always available. The so-called direct methods have provided viable solutions for the limited space of small molecules that diffract to near-atomic resolution [10]. Such methods could work for our protein fragments—*although they have higher resolution limits than preferred for such methods*. Our primary goal in this study is to establish a proof-of-concept for our novel approach combining deep learning with Patterson maps.

Within a broader scope, methods based solely on intensity measurements have been explored [11–13]. However, the discrete nature of crystallographic diffraction patterns poses significant challenges, limiting the applicability of algorithms such as the Gerchberg–Saxton and Fienup iterative methods, which are more suited to settings with continuous sampling conditions [14, 15].

Initial explorations into ML applications within crystallography include work by [16], who directly solve the phase problem for small organic molecules using a convolutional network to predict phases of a set of structure factors, given the corresponding amplitudes and template phases; they also use a "recycling" training procedure for iteratively improving their template inputs. Furthermore, Taniai et al. demonstrate the potential of transformer-based models to predict properties of crystalline structures by treating atoms as individual tokens [17]. This approach enables efficient attention mechanisms across (effectively infinitely) repeating unit cells, optimizing the prediction of various properties. Cao et al. utilize a transformer-based generative model for creating novel crystal structures within specified space groups, showcasing the versatility of ML in generating valid structural predictions [18]. Overall, due to the different problem setups and inputs between our work and the above, a comparison is not directly possible; we leave such a study open for future work.

## 3. `RecCrysFormer` **setup and architecture**

**Input/problem setup.** We posit that Patterson maps, when processed through a deep learning model, can effectively disclose the intrinsic atomic structure within a unit cell. We represent all electron density maps and Patterson maps using three-dimensional grids, enabling us to create tensor constructs that facilitate computational analysis.

To elucidate the underlying mathematics, consider the electron density map represented as a 3D array $\mathbf{e} \in \mathbb{R}^{N_1 \times N_2 \times N_3}$. The corresponding Patterson map, $\mathbf{p}$, shares the same dimensions as $\mathbf{e}$. In addition to the properties above, it can be shown to be derived through the following relationship:

$$\mathbf{p} = \Re\left(\mathcal{F}^{-1}\left(\mathcal{F}(\mathbf{e}) \odot \mathcal{F}(\widehat{\mathbf{e}})\right)\right) \approx \Re\left(\mathcal{F}^{-1}\left(|\mathcal{F}(\mathbf{e})|^2\right)\right), \tag{4}$$

where $\odot$ symbolizes element-wise multiplication of matrices. The operator $\mathcal{F}$ denotes the Fourier transform, and $\mathcal{F}^{-1}$ represents its inverse. Here, $|\mathcal{F}(\mathbf{e})|^2$ captures only the magnitude of the complex-valued Fourier transform of $\mathbf{e}$. Additionally, $\widehat{\mathbf{e}}$ refers to an inversed-shift version of $\mathbf{e}$, where each entry is defined as $\widehat{e}_{i,j,k} = e_{N_1-i, N_2-j, N_3-k}$, and $\Re$ emphasizes that the result is a real number.

**Training loss definition.** Representing our model by $g(\boldsymbol{\theta}, \cdot)$, our objective is to convert a Patterson map $\mathbf{p}$ into an estimate of the corresponding electron density map $\mathbf{e}$, framing our problem as a regression task. Considering a dataset $\mathcal{D}$ with pairs $\{\mathbf{p}_i, \mathbf{e}_i\}_{i=1}^n$, our training process seeks to optimize the parameters $\boldsymbol{\theta}$ by minimizing the loss function:

$$\boldsymbol{\theta}^\star = \arg\min_{\boldsymbol{\theta}} \left\{ \mathcal{L}(\boldsymbol{\theta}) := \frac{1}{n} \sum_{i=1}^n \|g(\boldsymbol{\theta}, \mathbf{p}_i) - \mathbf{e}_i\|_2^2 \right\}. \tag{5}$$

This mean squared error (MSE) is employed as the primary loss function $\mathcal{L}(\boldsymbol{\theta})$. Alternative metrics, tailored specifically to the nuances of crystallography and structural biology, can also be used in our training framework to promote specific properties; see below.

**Partial protein structures.** A fundamental step in protein structure determination involves leveraging the known primary sequence of amino acids. Furthermore, each proteinogenic amino acid's possible

3D electron density is well established, which facilitates the use of what we term as *standardized partial structures*. These partial structures represent the single most commonly occurring conformations of amino acids. Larger amino acids, due to their complex nature, exhibit a broader spectrum of possible conformations known as rotamers. Despite this variability, the predominant conformation for each amino acid has been experimentally determined and can be utilized to enhance prediction accuracy.[1]

For our machine learning model, denoted by $g(\boldsymbol{\theta}, \cdot)$, we incorporate these electron density maps of standardized partial structures into the training process. Specifically, let $\mathbf{u}_i^j \in \mathbb{R}^{M_1 \times M_2 \times M_3}$ represent the electron density map of the $j$-th amino acid in the $i$-th protein example. The map is centered by the center of mass within the unit cell. Our objective is now to optimize the model parameters $\boldsymbol{\theta}$ by minimizing the following loss function:

$$\boldsymbol{\theta}^\star = \arg\min_{\boldsymbol{\theta}} \left\{ \mathcal{L}(\boldsymbol{\theta}) := \frac{1}{n} \sum_{i=1}^n \| g(\boldsymbol{\theta}, \mathbf{p}_i, \mathbf{u}_i^j) - \mathbf{e}_i \|_2^2 \right\}. \tag{6}$$

Each protein segment example's number of *partial structures* corresponds directly to the number of amino acid residues present; this is maintained throughout both training and inference phases.

| **Patterson maps processing** | **Partial structures processing** |
|---|---|
| $\mathbf{X}^0 = \texttt{3DCNN}_{\mathbf{W}_c}(\mathbf{p}) \in \mathbb{R}^{c \times N_1 \times N_2 \times N_3}$ | $\mathbf{U}^j = \texttt{3DCNN}_{\mathbf{W}_u}(\mathbf{u}^j) \in \mathbb{R}^{c' \times M_1 \times M_2 \times M_3}$ |
| $\mathbf{X}^0 = \texttt{Partition}(\mathbf{X}^0) \in \mathbb{R}^{\frac{N_1 N_2 N_3}{d_1 d_2 d_3} \times c \times d_1 \times d_2 \times d_3}$ | $\mathbf{U}^j = \texttt{Partition}(\mathbf{U}^j) \in \mathbb{R}^{\frac{M_1 M_2 M_3}{d_1 d_2 d_3} \times c' \times d_1 \times d_2 \times d_3}$ |
| $\mathbf{X}^0 = \texttt{Flatten}(\mathbf{X}^0) \in \mathbb{R}^{\frac{N_1 N_2 N_3}{d_1 d_2 d_3} \times (c d_1 d_2 d_3)}$ | $\mathbf{U}^j = \texttt{Flatten}(\mathbf{U}^j) \in \mathbb{R}^{\frac{M_1 M_2 M_3}{d_1 d_2 d_3} \times (c' d_1 d_2 d_3)}$ |
| $\mathbf{X}^0 = \texttt{MLP}_{\mathbf{W}_c}(\mathbf{X}^0) \in \mathbb{R}^{\frac{N_1 N_2 N_3}{d_1 d_2 d_3} \times d_t}$ | $\mathbf{U}^j = \texttt{MLP}_{\mathbf{W}_u}(\mathbf{U}^j) \in \mathbb{R}^{\frac{M_1 M_2 M_3}{d_1 d_2 d_3} \times d_t}$ |
| $\mathbf{X}^0 \mathrel{+}= \texttt{PosEmbedding}(\frac{N_1 N_2 N_3}{d_1 d_2 d_3})$ | $\mathbf{U}^j \mathrel{+}= \texttt{PosEmbedding}(\frac{M_1 M_2 M_3}{d_1 d_2 d_3})$ |

Figure 2: Math representation of the preprocessing steps for Patterson maps and partial structures.

**Model architecture.** Inspired by the synergistic potential of Fourier transforms and self-attention within Transformer models [20], we introduce `RecCrysFormer`, an architecture combining 3D convolutional NNs (CNNs) and vision transformers. The approach leverages global contextual information from Patterson maps to predict electron density maps, employing a self-attention mechanism that integrates available partial protein structure data. `RecCrysFormer` incorporates 3D CNNs at both the initial and final stages, enveloping a Transformer core. Distinct convolutional paths process the Patterson map inputs and the additional partial structure electron density inputs, which originate from different domains (Patterson vs. direct).

–*Input Processing and Embedding.* A Patterson map input $\mathbf{p}_i \in \mathbb{R}^{1 \times N_1 \times N_2 \times N_3}$ is processed through a 3D CNN that applies "same" padding in order to maintain the spatial dimensions, and expands the number of feature channels. The output is then segmented into patches of size $c \times d_1 \times d_2 \times d_3$, where $c$ represents the number of channels, and $d_1, d_2, d_3$ are the spatial dimensions of each patch. These patches are subsequently flattened into one-dimensional "word tokens" of dimension $d_t$ via a Multi-Layer Perceptron (MLP), combined with learned positional embeddings, and fed into a custom multi-layer vision transformer.

For the partial structures $\mathbf{u}_i^j \in \mathbb{R}^{1 \times M_1 \times M_2 \times M_3}$, a similar process is followed using separate convolutional and patch-to-token embedding layers, producing additional tokens. The patch and token embedding operations for these inputs mirror those used for the Patterson map inputs, ensuring consistent data treatment across different input types; see Figure 2.

–*The core transformer.* `ResCrysFormer` integrates a novel attention mechanism tailored for enhancing the prediction of protein structures from 3D Patterson maps and partial structures. The core innovation lies in the interaction between the tokens derived from Patterson inputs or the previous transformer layer and the tokens derived from partial structures, which are concatenated before

---

[1]We obtain the atomic coordinates and derive the corresponding electron density maps for these conformations using the "Get Monomer" feature of the Coot program [19].

generating the "$Q, K, V$" matrices. Note that in our notation, these are created via the corresponding $\mathbf{W}_q^h, \mathbf{W}_k^h, \mathbf{W}_v^h$ trainable query, key, and value projection matrices of the $h$-th attention head for tokens from the Patterson map; $\mathbf{W}_{k'}^h, \mathbf{W}_{v'}^h$ are the corresponding matrices for partial structure tokens. Thus our "$Q$" matrix is effectively truncated compared to "$K$" and "$V$", so that only tokens derived from Patterson maps can attend tokens from the partial structures. This one-way attention reduces the computational costs of our model.

We do not generate new partial structure token embeddings in each transformer layer, but instead maintain a constant initial embedding for these tokens across all layers. Thus, each layer utilizes the electron density information present in our partial structures as a stable reference point for attention calculations. Formally, our attention mechanism can be described as follows; see also Figure 3:

$$\mathbf{U} = \texttt{Concat}_{j=1}^{J}(\mathbf{U}^j) \in \mathbb{R}^{(S' J) \times d_t},$$

$$\mathbf{A}^h = \texttt{Softmax}\left((\mathbf{W}_q^h \mathbf{X}^\ell)(\texttt{Conc.}(\mathbf{W}_k^h \mathbf{X}^\ell, \mathbf{W}_{k'}^h \mathbf{U}))^\top\right)$$

$$\widehat{\mathbf{V}}^h = \mathbf{A}^h \left(\texttt{Concat}(\mathbf{W}_v^h \mathbf{X}^\ell, \mathbf{W}_{v'}^h \mathbf{U})\right) \in \mathbb{R}^{S \times d_h},$$

$$\mathbf{O} = \mathbf{W}_o \texttt{Concat}\left(\widehat{\mathbf{V}}^0, \ldots, \widehat{\mathbf{V}}^{H-1}\right) \in \mathbb{R}^{S \times d_t},$$

$$\mathbf{X}^{\ell+1} = \mathbf{W}_{\text{ff2}}(\texttt{ReLU}(\mathbf{W}_{\text{ff1}} \mathbf{O})).$$

where the token sequence lengths for the Patterson and partial structure inputs are $S = \frac{N_1 N_2 N_3}{d_1 d_2 d_3}$ and $S' = \frac{M_1 M_2 M_3}{d_1 d_2 d_3}$, respectively. $d_h$ denotes the token embedding dimension after splitting into attention heads and $H$ the number of attention heads. $\mathbf{W}_{\text{ff1}}$ and $\mathbf{W}_{\text{ff2}}$ are the trainable parameters of the fully connected layers in a standard MLP feed-forward block.[2]

Figure 3: Overview of our transformer layer

*–3D Reconstruction Layers.* After processing through the transformer layers, the token representations are rearranged and then transformed back into a 3D electron density map using another series of 3D convolutional layers:

$$g(\boldsymbol{\theta}, \mathbf{p}) = \texttt{tanh}(\texttt{3DCNN}_{\mathbf{W}_o}(\texttt{Rearrange}(\texttt{MLP}(\mathbf{X}^L)))).$$

where $L$ is the number of transformer layers. These final transformations are critical for translating the learned abstract features back into a spatially coherent structural format.

**Enhanced loss definition.** We use a combination of the MSE loss and a few instances of the negative Pearson correlation coefficient. The Pearson correlation is an oft-used metric in crystallography that can be easily calculated between two densities. If we denote a model prediction as $\mathbf{e}'$, and define $\bar{\mathbf{e}} = \frac{1}{N_1 N_2 N_3} \sum_{i,j,k} \mathbf{e}_{i,j,k}$ and $\bar{\mathbf{e}}' = \frac{1}{N_1 N_2 N_3} \sum_{i,j,k} \mathbf{e}'_{i,j,k}$, then the Pearson correlation coefficient between $\mathbf{e}$ and $\mathbf{e}'$ is as below:

$$\texttt{PC}(\mathbf{e}, \mathbf{e}') = \frac{\sum\limits_{i,j,k=1}^{N_1,N_2,N_3} (\mathbf{e}'_{i,j,k} - \bar{\mathbf{e}}')(\mathbf{e}_{i,j,k} - \bar{\mathbf{e}})}{\sqrt{\sum\limits_{i,j,k=1}^{N_1,N_2,N_3} (\mathbf{e}'_{i,j,k} - \bar{\mathbf{e}}')^2} \cdot \sqrt{\sum\limits_{i,j,k=1}^{N_1,N_2,N_3} (\mathbf{e}_{i,j,k} - \bar{\mathbf{e}})^2}}, \tag{7}$$

Since a larger Pearson correlation indicates a more accurate prediction, we take negations in order to use these correlations as additional loss function terms. One Pearson term in our loss function is obtained by directly taking the negative Pearson correlation between model predictions and ground truth, and the other by taking the negative Pearson after first applying a Fourier transform to both the model prediction and ground truth, and then taking the amplitudes of all elements in the resulting complex tensors. The overall loss function then becomes:

$$\mathcal{L}(\boldsymbol{\theta}) := (\frac{c_{\text{MSE}}}{n} \sum_{i=1}^{n} \|\mathbf{e}'_i - \mathbf{e}_i\|_2^2) - (\frac{c_P}{n} \sum_{i=1}^{n} \texttt{PC}(\mathbf{e}, \mathbf{e}')) - (\frac{c_P}{n} \sum_{i=1}^{n} \texttt{PC}(|(\text{FFT}(\mathbf{e})|, |\text{FFT}(\mathbf{e}')|)). \tag{8}$$

where $c_{\text{MSE}}$ and $c_P$ denote the relative weights of the MSE and negative Pearson components.

---

[2]We omit skip connections, layer normalization, and attention scaling factor to simplify notation; these are included in practice.

# 4. Recycling meta-algorithm

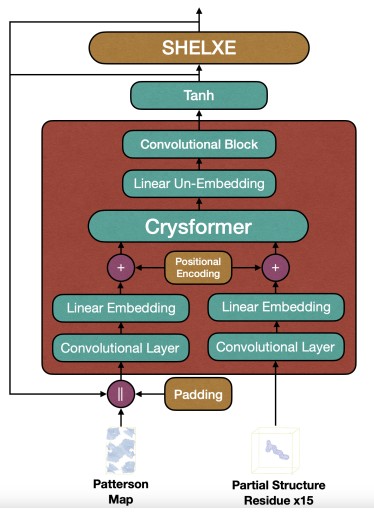

Figure 4: `RecCrysFormer` meta-algorithm. Arrows show the information flow among the various components.

`RecCrysFormer` incorporates a custom recycling training process, which leverages existing predictions of electron densities as input into a crystallographic refinement program that produces atomic structure estimates. These refined estimates are then re-utilized to generate improved electron density maps, creating a cyclic enhancement in model accuracy. This concept is inspired by the iterative refinement approaches used in notable studies such as AlphaFold2 [3, 21]; see Figure 4.

**Implementation details.** Initially, our model employs a standard training run to generate electron density maps from given Patterson maps and partial structures. In subsequent recycling training iterations, each of which performs a complete end-to-end training, the architecture remains unchanged except for an augmentation in the first convolutional layer to accept an additional input channel. *This channel introduces a template electron density map, which serves as a guide that directs the model toward more accurate structural reconstructions.* Such a procedure resembles ideas from data augmentation techniques in traditional and adversarial ML training [22, 23]

The templates for recycling are primarily derived from the outputs of the initial training phase or the most recent recycling iteration. For enhanced accuracy, some templates are replaced with refined maps obtained via a process based on the SHELXE program [24, 25], significantly enriching the training data diversity. This is a separate procedure that is not part of the ML model, being performed in between training runs. After the initial training run, about $92\%$ of the training set and $79\%$ of the test examples were able to be refined with SHELXE.

**Challenges and adaptations.** Originally, each model prediction template had a $70\%$ chance of being substituted with a refined template if available during training. While this yielded substantial improvements, it often led to overfitting on these refined templates. This created better accuracy for well-predicted initial structures, but poorer outcomes for less accurately predicted initial structures.

To address this, we adapted our recycling approach in the following ways: $i$) we greatly reduced the frequency of refined template usage during training to $1/6$, reducing overfitting while still benefiting from their accuracy; $ii$) we introduced Gaussian noise to the template input channel during training to promote robustness against minor input variations and further develop our data augmentation and adversarial training aspects—this does not apply to the Patterson map channel to maintain structural integrity; $iii$) we employed transfer learning techniques by initializing the model weights from the most recently trained model (after applying slight Gaussian noise). We transferred only half of the weights in the first convolutional layer for the first recycling iteration to balance between learned weights on the Patterson map inputs and necessary randomization for weights acting on the previously unseen template map inputs.

# 5. Experimental Methods and Results

**Model implementation details.** Due to resource constraints, we made use of a slightly modified version of the Nyström approximate attention procedure [26] instead of classical self-attention for our training. This means that in theory, our entire model scales linearly with the number of elements in our input tensors. Our partitioning and flattening operations can be performed simultaneously in one layer, and we use a single linear layer followed by partitioning to generate our query, key, and value matrices, followed by a truncation of the query matrix across the sequence length dimension. The first 3DCNN component is a single layer with kernel size 7. In contrast, the second 3DCNN post-

| Metric | Initial | Recycling | Recycling (modified) |
|---|---|---|---|
| Mean PC($\mathbf{e}, \mathbf{e}'$) | 0.817 | $\mathbf{0.929}_{\uparrow(13.7\%)}$ | $\mathbf{0.918}_{\uparrow(12.3\%)}$ |
| Mean PC($\mathbf{e}, \mathbf{e}'$) (not refined) | 0.658 | $0.670_{\uparrow(1.8\%)}$ | $\mathbf{0.738}_{\uparrow(12.1\%)}$ |
| Mean Phase Error | 64.3° | $\mathbf{20.2°}$ | 33.8° |
| Mean Phase Error (not refined) | 79.6° | 76.4° | $\mathbf{64.5°}$ |
| Percent Refined | 79 | 87 | 93 |
| Epochs | 110 | 80 | 80 |

Table 1: Comparison of training runs. Certain rows (2, 4) report results only for the $\sim 21\%$ of test set examples for which the prediction after the initial run could not be refined with SHELXE.

transformer component is a sequence of two residual blocks with layers of kernel size 5, taken from the BigGAN [27] architecture, followed by a final convolution with kernel size 3. As stated, we apply "same" padding in all convolutional layers as our input Patterson and output electron density map have the exact same shape for each example. We use the circular padding scheme in our initial 3DCNN component due to the inherent periodicity of our Patterson maps and partial structures.

**Optimizer and training details.** Training was performed using the AdamW optimizer [28], with the OneCycle learning rate schedule [29]. Due to a considerable difference in magnitude between values of the negative Pearson and MSE during training, and to promote the well-established MSE being the main driving force of our training, we weighed the MSE by 0.9999 and the two negative Pearson correlation terms by $5 \times 10^{-5}$.

A table reporting the model hyperparameters we used for our training runs on our 15-residue variable unit cell and angle dataset can be found in the appendices, where $d_{ff}$ refers to the hidden dimensionality within our feed-forward MLP blocks. All such training runs were performed on a single RTX 6000 Ada GPU with 48 GiB memory, with torch.set_float32_matmul_precision set to 'high'. On average, one training epoch required about 314 minutes for our initial training run and about 318 minutes for our recycling runs. Due to time limitations, we have only performed one instance of our initial, recycling, and "modified" recycling training runs.

**Metrics.** For model evaluation, we continue to use the Pearson correlation coefficient between our ground truth targets $\mathbf{e}$ and model predictions $\mathbf{e}'$. Additionally, we conduct phase error analysis on structure factors derived from our models' final predictions following our training runs; phase error is generally considered a more informative metric than Pearson correlation This is done via the cphasematch tool from the CCP4 software suite [30], which reports the mean phase errors of our predictions' structure factors in degrees across various ranges of reflection resolution, with lower phase errors indicating more accurate phases.

**Evaluation Results** The bulk of the results presented below were obtained on a dataset of examples generated with a constant grid sampling rate and a constant resolution limit of 1.5 Å; see the appendices for a detailed description of our data generation process. This dataset consisted of $348,880$ training and $38,291$ test examples for a roughly $90\% - 10\%$ split. We consider the following training regimes for comparison:

- Initial: Our very first training run, with no template map inputs provided.

- Recycling: Recycling run using our original recycling formulation, where a model prediction (produced from the final state of the Initial run) template map has a $70\%$ chance of being substituted with the corresponding SHELXE-refined map, if available, during training.

- Recycling (modified): Recycling run with the same inputs as above, but using our modified algorithm with a lower chance of substitution with refined map, addition of Gaussian noise, and transfer learning from the final model state of the Initial run.

Evaluation results on our test set after each of the three defined training runs are shown in Table 1; phase error is reported as an average across all ranges of resolution. During evaluation after our recycling runs, we do not set a chance for a model prediction template to be replaced by the corresponding SHELXE post-refinement map; instead, we replace all templates for examples successfully refined after the initial training run ($\sim 79\%$ of the test set).

All metrics were greatly improved after our recycling runs compared to after our initial training run, as expected. See the first row of Table 1, where both `Recycling` and `Recycling (modified)` achieve $> 10\%$ improvement over the vanilla implementation of our methodology. When comparing our results from our modified recycling training regimen to our original recycling formulation, we also find that our modified process further improves predictions for the failure case examples that had been unable to be refined in SHELXE after the initial training run (and thus always has the model prediction from the end of the initial training as the template map); see rows 2 and 4.

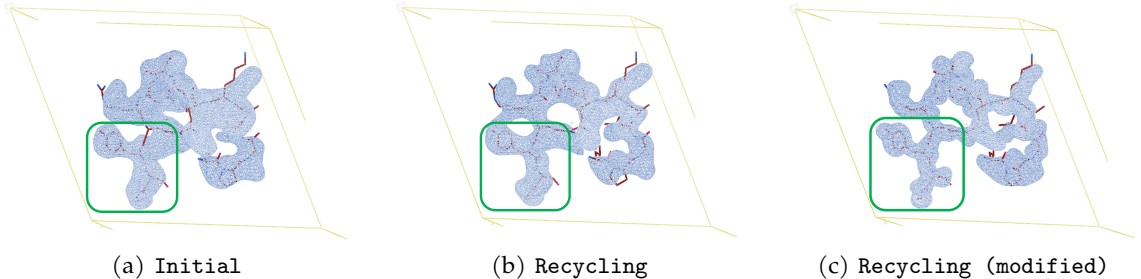

(a) Initial        (b) Recycling        (c) Recycling (modified)

Figure 5: A test set example (4AZ3_1.pd_11) representing the failure case where SHELXE could not produce a refined map. The underlying ground truth model is shown in red. Our first recycling formulation only slightly improves most aspects, but the prediction after our modified run shows clear improvement in several details. See the highlighted box for a region that demonstrates this.

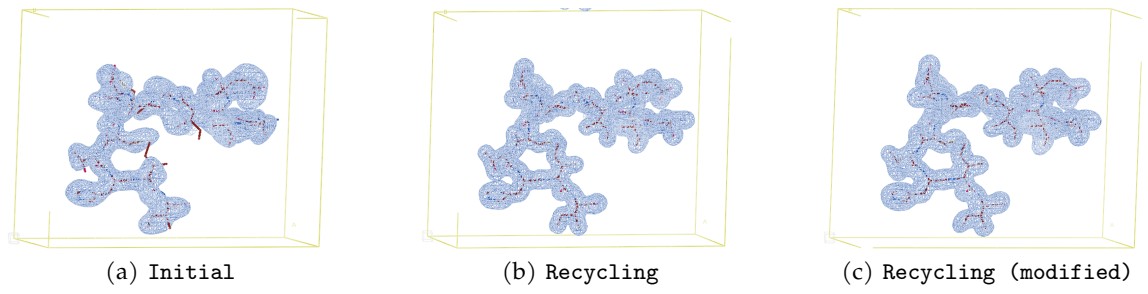

(a) Initial        (b) Recycling        (c) Recycling (modified)

Figure 6: A test set example (3W4R_1.pd_21) that can be refined after the initial training run. The initial prediction is already reasonable, but the prediction after both recycling runs almost exactly matches the underlying atomic coordinates in all aspects.

On the other hand, predictions for the examples that were able to be successfully refined after the initial training run tended to be slightly worse than before. For this dataset, the large majority of test examples were successfully refined after the initial run, so we found that results across the entire test set were slightly worse than with our original recycling formulation (but still highly successful overall). We consider this tradeoff worthwhile, especially for more complex future datasets where we expect to obtain worse predictions after an initial training run.

We visualize some model predictions on our test set in Figures 5, 6. Predicted densities are shown in blue, while the ground truth atomic model is shown in stick representation.

**Phase error analysis.** We report the fraction of predictions with less than $60°$ mean phase error at various ranges of reflection resolution in Figure

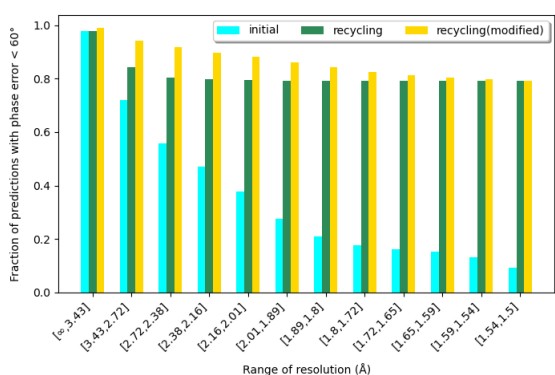

Figure 7: Fraction of test set predictions with phase error $< 60°$ at different ranges of reflection resolution. By convention, lower resolution ranges are towards the left.

7. We chose this value as, in general, if the phase error for the reflections at a particular resolution range is less than $\sim 60°$, the corresponding phases are considered solved. Reflections at lower resolution generally indicate the overall shape of an underlying atomic structure, while those at higher resolution indicate finer structure details [31].

Our results show that the predictions after the initial training run match the low-level shape of the desired underlying structures well but can be inaccurate in the exact details. Meanwhile, predictions after both recycling training runs are much more accurate in reproducing details, as shown by the visualizations in Figures 5, 6. And even if most of the test predictions had slightly higher mean phase errors across all resolution ranges after our modified recycling regimen, they still were always below our $60°$ cutoff for successful phases. Combined with the improvements in predictions for the minority of examples for which refined maps were not obtained after the initial training run, this led to an improvement in phase error results at almost all resolution ranges.

**Variable resolution dataset** Although we consider a promising step towards a solution for real-world structures, unrealistic aspects such as their high solvent content, small size, and unusually high resolution limit mean that existing direct methods can also be applied. E.g., running the SHELXD dual-space direct method [32] at 1.5 Å on our test set examples resulted in a $97\%$ success rate. Thus, we also provide results after initial and (modified) recycling training runs on a slightly larger dataset of $384,814$ training and $41,774$ test examples, with the same number of residues but a variable grid sampling rate and variable resolution limit in the range of 1.6 Å to 2.25 Å (average of 2.0 Å), in Table 2. This is a harder problem, and running SHELXD on the test set examples of this dataset at 2.0 Å resulted in a success rate of only $73\%$. In contrast, after one recycling training run, our method is able to outperform SHELXD. We will further modify our recycling training procedure to obtain better results as future work.

| Metric | Initial | Recycling (modified) |
|---|---|---|
| Mean $\text{PC}(\mathbf{e}, \mathbf{e}')$ | 0.76 | 0.854 |
| Mean $\text{PC}(\mathbf{e}, \mathbf{e}')$ (not refined) | 0.644 | 0.676 |
| Mean Phase Error | $65.2°$ | $35.3°$ |
| Mean Phase Error (not refined) | $76.7°$ | $68.2°$ |
| Percent Refined | 64 | 83 |
| Epochs | 110 | 80 |

Table 2: Results on variable resolution and grid sampling dataset. Certain rows report results only for the $\sim 36\%$ of test set examples for which the prediction after the initial run could not be refined.

## 6. Conclusions, Limitations and Future Work

This work represents a step in developing a general direct translation from experimental crystallographic data to electron density estimates, and thus 3D atomic structures of proteins, which due to the crystallographic phase problem existing literature has not yet been able to solve. We have effectively established the fundamental capability of our model to learn the relationship between Patterson maps and electron densities on small synthetic structures, and identified various architectural components, training strategies, and integration of crystallographic techniques that enable this.

**Limitations.** Our examples are still smaller than actual proteins and have a high solvent content (i.e., amount of space) in the unit cell, and so we will need to systematically increase the segment lengths of our examples in order to assess scalability towards full-length proteins. Other potential difficulties include experimental noise and a substantial amount of peak overlaps in the Patterson map.

**Future Work.** We have not yet considered internal symmetry within unit cells. Thus, all of our examples are of the $P1$ space group, but there are 64 other possible space groups for proteins, each with some form of internal symmetry. In future work, we will train our model on datasets of examples belonging to one of several possible space groups; this will make our examples contain more than one molecule per unit cell. We will also consider multiple different choices of unit cells and space groups for the same crystal structure. Addressing our current limitations will require synthesizing datasets with a larger number of total examples, and adjusting hyperparameters to increase model

complexity. Due to the expected increase in time and space training loads, we will look into methods that improve the efficiency of our model, such as downsampling within the transformer.

## Acknowledgements

This research was funded in part by: The Robert A. Welch Foundation (grant No. C-2118 to G.N.P and A.K.); NSF, Directorate for Biological Sciences (grant No. 1231306 to G.N.P.); Rice University (Faculty Initiative award to G.N.P and A.K.); NSF FET:Small (award no. 1907936); NSF MLWiNS CNS (award no. 2003137, in collaboration with Intel); NSF CMMI (award no. 2037545); NSF CAREER (award no. 2145629); a Rice InterDisciplinary Excellence Award (IDEA); an Amazon Research Award; a Microsoft Research Award. AK would also like to thank the Ken Kennedy Institute for its support through the Research Cluster "AI-OWLS". The content is solely the responsibility of the authors and does not necessarily represent the official views of the Funders.

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

# A. Appendix

# B. Data Generation Process

We have devised a standardized methodology to generate datasets composed of synthetic short protein fragments extracted from entries in the Protein Data Bank (PDB) [33]. Starting with a curated set of protein structures from the PDB, we extract all segments of adjacent amino acid residues of a

specified length from these structures. Specifically, the dataset introduced in this study has segments of 15 residues. The generated Patterson and electron density maps have the same dimensionality for each example. Furthermore, the examples are placed in unit cells of variable axis lengths and angles. We also use a smaller and simpler dataset of 2-residue segments for our hyperparameter studies.

**Structure Curation from the Protein Data Bank.** We start with a curated set of approximately 24,000 structures from the PDB that satisfy the following criteria: proteins solved by X-ray crystallography after 1995 with sequence length $\geq 40$, refinement resolution $\leq 2.75$, and R-Free $\leq 0.28$, and with clustering at 30% sequence identity and available in legacy PDB format. From these, we extract segments of adjacent amino acid residues of a specified length. For our dataset with 15-residue segments, we allow potential overlaps of up to 5 residues between consecutive segments in order to increase the starting amount of possible examples. To prevent overfitting effects that could result from this overlap of subsegments, all examples derived from the same original PDB structure are placed together in either the training or test set.

**Preprocessing and Validation.** Using the `pdbfixer` Python API [34], we filter out all segments that contain non-standard residues, missing residues, or missing atoms. We apply the following standardized modifications to the remaining viable examples to enhance the consistency and reliability of our dataset: set all temperature factors to 20, rebuild all selenomethionine residues as methionine, and remove all hydrogen atoms.

**Unit Cell Configuration.** Each protein fragment now undergoes an iterative process to define its unit cell:

—We begin by determining the raw $\mathrm{max} - \mathrm{min}$ ranges of Cartesian coordinates along each axis.

—We iteratively increase the current unit cell dimensions, intending to ensure a minimum intermolecular atomic contact distance of greater than 3.45 Å. [3]

—The three angles of the unit cell are randomly set to $90°, 100°, 110°,$ or $120°$, with respective selection probabilities of $1/3, 1/3, 2/9,$ and $1/9$.

—Fragments still with intermolecular atomic clashes below 3.45 Å are discarded.

—A reindexing operation ensures the longest and shortest axes are consistently oriented.

To address the issue of translation invariance in Patterson maps, which could lead to ambiguities in training [35], we adjust all atomic coordinates so the center of mass is at the unit cell's center. This strategic placement aids in maintaining the predictability and consistency of the electron densities within their respective unit cells, as model predictions and desired densities will always be located towards the center of the unit cell. Furthermore, this is theoretically justifiable as unit cell boundaries relative to the contents thereof are essentially arbitrary.

**Centrosymmetry-induced Patterson map ambiguity** It is known that an atomic structure and its centrosymmetry-related structure will always exhibit the same Patterson map, leading to another potential ambiguity in Patterson map interpretation. Previous work [35] proposed to solve this issue by always combining a set of atoms with their centrosymmetry-related counterparts into one example, which necessitates a post-processing algorithm to differentiate between the original and centrosymmetric densities for each model prediction. To avoid this, we leverage the inherent characteristics of real protein structures. Specifically, proteinogenic amino acids naturally occur only in a single mirror-image symmetry (enantiomer) configuration [36]. While mirror-image symmetry fundamentally differs from centrosymmetry, we have found that so far, this consistency in our example structures enables us to train on unmodified Patterson and electron density maps.

**Map Generation and Normalization.** Structure factors are computed for each protein fragment using the `gemmi sfcalc` program [37], up to a maximum resolution of either a constant 1.5 Å (for our constant resolution and grid spacing dataset), or a variable resolution in the range of 1.6 Å to 2.25 Å

---

[3]An Angstrom (Å) is a metric unit of length equal to $10^{-10}$ meters.

(for our variable resolution and grid spacing dataset). From these structure factors, electron density and Patterson maps are derived using the fft program from the CCP4 suite [38, 39]. These maps are sampled at either a rate of 3.0, resulting in a grid spacing of 0.5 Å, calculated as $1.5Å/3.0 = 0.5Å$ (constant dataset), or a rate in the range of 2.305 to 2.745 (variable dataset). All maps are then converted into PyTorch tensors. The tensor values are normalized to the range $[-1, 1]$ to accommodate the use of a tanh final activation layer in our models. To ensure uniform shape across the examples in our training batches as required by PyTorch, examples that belong to tensor-size bins smaller than our minimum batch size are excluded from the training set.

## C. Nomenclature of our Example IDs

Each of our training and test set examples has an associated identifier consisting of three distinct sections delimited by underscore characters. The initial section is the PDBid of the protein structure from which the example was derived. The second denotes the specific entity of the aforementioned PDB structure from which the example was derived. The third section indicates the relative position of the first amino acid residue present in the example if all absent or unmodelled residues in the aforementioned PDB entity were excluded.

## D. Links to dataset and code base

We provide intermediate files from our process for generating our constant resolution and grid spacing dataset, and a subset of our data generation and training codebase sufficient to create our tensor inputs and outputs, and reproduce our results on this dataset. Due to space constraints, we divide this into separate Zenodo datasets. The working directory can be extracted from: https://doi.org/10.5281/zenodo.11244967. See the README.md file in the directory for instructions on running our scripts. Additional files are found at the following links: https://doi.org/10.5281/zenodo.11239133, https://doi.org/10.5281/zenodo.11239205, https://doi.org/10.5281/zenodo.11239285, https://doi.org/10.5281/zenodo.11239432, https://doi.org/10.5281/zenodo.11239765. We alternatively provide the codebase at https://github.com/sciadopitys/RecCrysFormer, into which the additional files can be extracted after download.

## E. Phase refinement with SHELXE

We used the standard crystallographic phasing program SHELXE[4, 25] both for the evaluation of the predicted electron density maps and to prepare refined maps for use in our training recycling. This method uses density modification to improve the phases during fitting and results in poly-alanine backbone models. For each test case, four global tracing cycles were run with each global cycle having ten rounds of density modification. Density modification involved applying physical constraints in real space and iterative projection back to reciprocal space. The constraints applied included negative density truncation in the protein region and density flipping in the solvent with a weighted combination for voxels with intermediate scores for protein versus solvent identified using the sphere-of-influence method [4].

To evaluate the maps, the SHLEXE program iteratively interprets the maps by building poly-alanine molecular fragments into the electron density map [24, 25]. It assesses the quality of the poly-alanine model by calculating the structure factor amplitudes from the model. When the Pearson correlation coefficient of the model amplitudes with the true underlying structure factor amplitudes exceeds 0.2, then the model is highly likely to be successfully refined.

When using the maps produced from the SHELXE procedure in recycling training, we found it best to further flatten the solvent region. We did this by making two masks, one with all positive values from the SHELXE map and the other after blurring the SHELXE map with a crystallographic B factor of 25 and selecting all voxels with a density greater than 1.25 R.M.S.D above the map mean. These

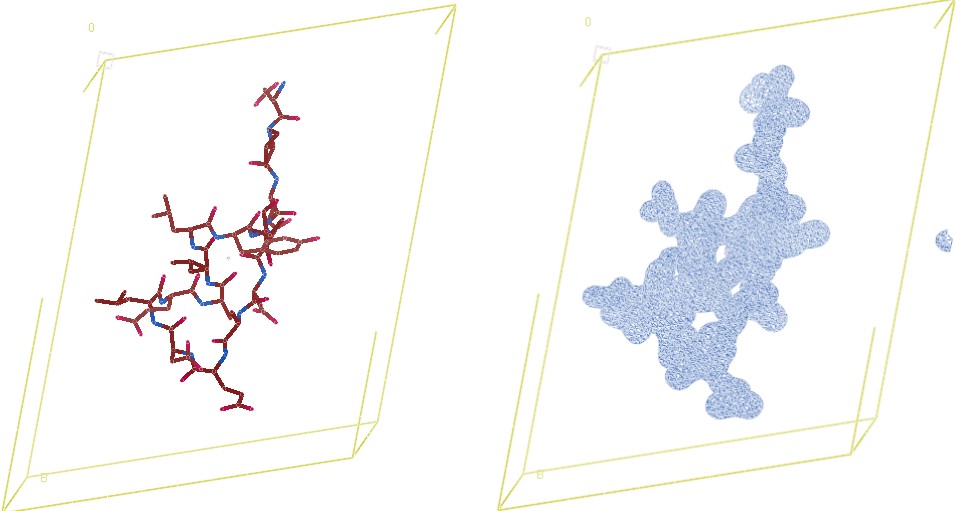

Figure 8: A generated 15-residue protein fragment within a unit cell of variable axis lengths and angles (left) and a view of the corresponding electron density (right).

masks were applied by adding 1.23 to the SHELXE map, multiplying the result by the two masks, and subtracting 1.23 from the result. This new map was then combined with the original SHELXE map using a weighting of (0.95*modified + 0,05* raw). The resulting map was then scaled to the range -1 to 1 and stored in a PyTorch tensor.

## F. Additional visualizations of model predictions

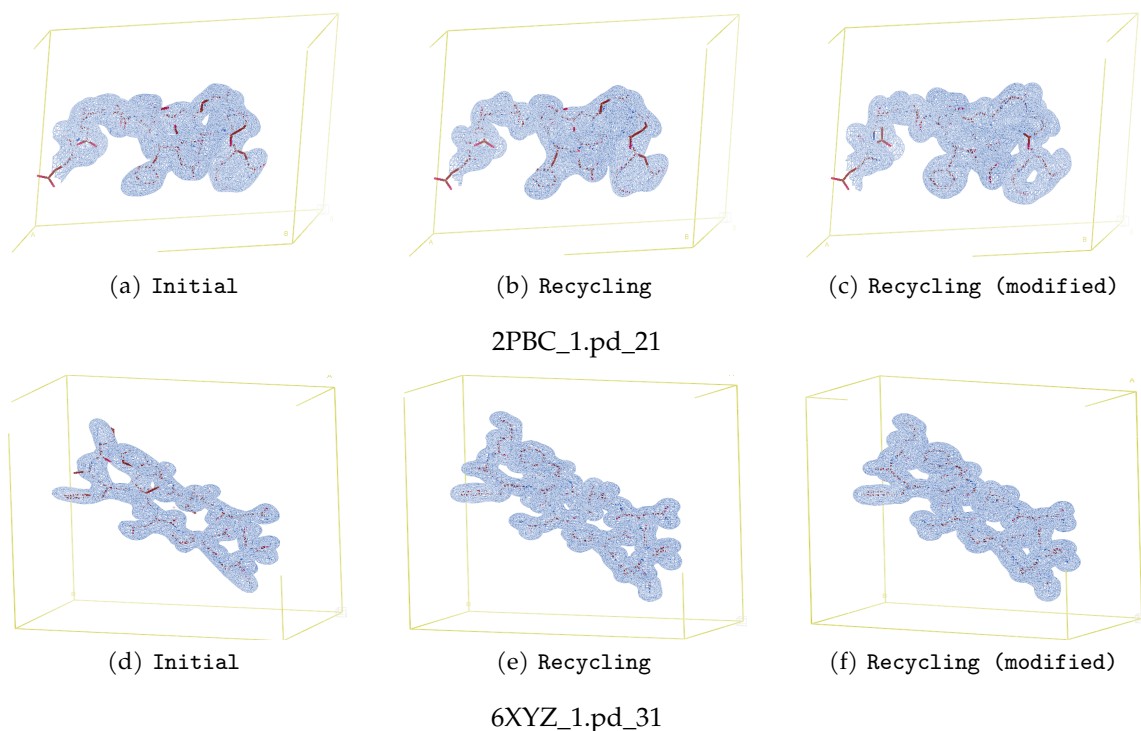

(a) Initial          (b) Recycling          (c) Recycling (modified)

2PBC_1.pd_21

(d) Initial          (e) Recycling          (f) Recycling (modified)

6XYZ_1.pd_31

# G. Hyperparameters for Training and Evaluation runs

| Hyperparameter | Value |
|---|---|
| $c = c'$ | 10 |
| $d_1 = d_2 = d_3$ | 4 |
| $d_t$ | 512 |
| $d_h$ | 64 |
| $H$ | 8 |
| $L$ | 10 |
| $d_{ff}$ | 2048 |
| AdamW weight decay | $3e-2$ |
| starting lr | $5e-4$ (initial), $2.5e-4$ (recycling) |
| max lr | $1.4e-3$ (initial), $8e-4$ (recycling) |
| average batch size | 40 |

# H. Approximate Hyperparameter Tuning with Dipeptide Dataset

The `RecCrysFormer` has shown success in predicting complicated 15-residues examples. However, the size of the `RecCrysFormer` is large enough that it is prohibitively time-consuming to conduct ablation studies to tune hyperparameter values. Instead, we ran a scaled down version of the `RecCrysFormer` model on a smaller and easier dipeptide dataset for a shortened number of epochs. Drawing intuition from these results, we were able to identify more promising ranges for the `RecCrysFormer` hyperparameters. The dipeptide model had default parameters set to 3; which produced an average Pearson correlation of $0.723$. At each trial, we varied a parameter. Selected results of these experiments are included in H.

| | |
|---|---|
| Embedding Dimensions | 512 |
| Crysformer MLP Dimensions | 512 |
| Attention Head Count | 8 |
| Patch Size | 4 |
| Attention Block Depth | 6 |

Table 3: Default hyperparameter values for tuning on dipeptide dataset.

| Parameter Changed | Value | Pearson Correlation | Parameter Changed | Value | Pearson Correlation |
|---|---|---|---|---|---|
| Head Count ($H$) | 2 | 0.684 | Depth ($L$) | 4 | 0.700 |
| | 4 | 0.696 | | 4 | 0.702 |
| | 4 | 0.731 | | 4 | 0.723 |
| | 4 | 0.707 | | 8 | 0.746 |
| | 16 | 0.751 | | 16 | 0.771 |
| | 32 | 0.765 | | 16 | 0.792 |
| | 32 | 0.757 | MLP Dim ($d_{ff}$) | 32 | 0.706 |
| | 32 | 0.767 | | 64 | 0.696 |
| | 64 | 0.742 | | 64 | 0.715 |
| Embedding Dim ($d_t$) | 32 | 0.641 | | 128 | 0.713 |
| | 64 | 0.684 | | 256 | 0.725 |
| | 256 | 0.706 | | 256 | 0.731 |
| | 256 | 0.703 | Residual Block Size | 0 | 0.713 |
| Patch Size ($d_1 = d_2 = d_3$) | 8 | 0.696 | | 0 | 0.717 |

Table 4: Selected results of hyperparameter tuning trials on the dipeptide dataset.

