# OpenReview forum: "RecCrysFormer: Refined Protein Structural Prediction from 3D Patterson Maps via Recycling Training Runs"
_CPAL.cc/2025/Proceedings_Track — CPAL 2025 (Proceedings Track) Poster_

### Official Review · Reviewer_SA1i · 2025-01-10
**Combine transformer and 3D convolutional network for electron density maps prediction.**

**Rating:** 6
**Confidence:** 4

**Review:**

**Summary**

This paper presents RecCrysFormer, a hybrid deep learning framework that combines transformers and 3D convolutional networks to predict protein electron density maps directly from Patterson maps. The authors highlight a “recycling” training regimen that iteratively refines outputs using crystallographic tools (e.g., SHELXE) and template electron densities, leading to improved accuracy across multiple metrics. Leveraging partial protein structures in standard conformations further refines the network’s predictions. Experiments on synthetic peptide fragments with variable unit cell parameters demonstrate the model’s robustness. This approach seeks to close the gap between classic crystallographic methods and modern ML-based structure prediction, potentially benefiting real-world crystallographic pipelines.

**Strengths**

1. The paper’s core idea of merging 3D CNNs, transformers, and partial amino-acid conformations for electron density prediction from Patterson maps is creative.
2. Iterative “Recycling”: The meta-algorithm that reintroduces refined outputs into subsequent training adds a practical feedback loop, reminiscent of advanced techniques in protein structure prediction.
3. Robust Experimental Setup: The preliminary experiments on synthetic data with different cell sizes and angles illustrate the method’s robustness and offer promising directions for future real-case scenarios.

**Weaknesses**

1. It is unclear how each component (e.g., partial structures, specific attention mechanisms) contributes to performance without a thorough ablation study or comparison to baseline models.
2. Limited Interpretability: Despite its sophisticated architecture, the paper does not offer intuitive explanations or strong empirical evidence that clarifies why the model works so well.
3. Absence of Larger Protein Experiments: The experiments focus on small fragments, so scalability to bigger proteins remains unproven.
More insight into potential pitfalls with noisy or incomplete Patterson maps would strengthen the claims.

**Questions**

1. Multiple versions of the loss function are mentioned, including MSE, negative Pearson correlation, and their combinations. Could you clarify the distinctions among these definitions and specify exactly which loss formulation was ultimately used in the experiments?
2. You state that the number of partial structures used matches the number of residues in each protein fragment. Could you explain in greater detail why these counts must be equivalent and how partial structures are integrated (or excluded) if non-standard residues or alternate conformations exist?
3. Could you provide more details on computational overhead? Specifically, how does the model’s run-time and memory usage scale when increasing the size of the input Patterson maps or the length of the protein segments?

---

### Official Review · Reviewer_tEVw · 2025-01-10
**Review of RecCyrsFormer: Refined Protein Structural Prediction from 3D Patterson Maps via Recycling Training Runs**

**Rating:** 6
**Confidence:** 2

**Review:**

RecCrysFormer is an approach to determine structure of proteins using a CNN/ViT architecutre that predicts electron density maps from Patterson maps.

Strengths
1. The approach seems novel from my limited knowledge of the related literature. The authors state that this approach using Patterson maps is new. The architecture is well-motivated and utilizes structure specific to the protein problem.
2. The experiments showcase the recycling procedure works quite well at the given metrics.

Weaknesses
1. All equations should be numbered so that they can be referred to easily by reviewers and readers.
2. Is there a need to regularize the solution the objective in Line 138 - recovering the density structure from Patterson map resembles a phase retrieval inverse problem, so there can be multiple possible solutions. Typically, these inverse problems are solved by imposing some natural structure on the recovered point e.g sparsity, lying in the range of generative model, etc. Is that the role of the partial structures - I did not quite understand what these partial structures are. How would this compare to a more standard regularizer for the phase retrieval problem?
3. The paper should describe the training procedure - namely, in the recycling procedure, is everything trained end to end? If so, is there differentiation done through the SHELXE procedure? The paper did not explain how training is done - is the crysformer architecture initialized at random or some pretrained model, and then how is training done?
4. As someone not an expert in protein structures, I find Figure 5 quite hard to discern what is good and what is bad from the results? A, B and C all look pretty similar and the red box does not quite help determine what the reader should be focusing on. An explanation in words would help. Is there a ground truth to compare to?

Questions
1. What is script R in Line 115?
2. What is nystrom attention - is that the proposed attention?

Overall, the paper seems well-written and the results are compelling. Given the weaknesses above, I recommend 6 although I am not that confident in these weaknesses.

---

### Official Review · Reviewer_vVRx · 2025-01-13
**Review of paper Submission43**

**Rating:** 7
**Confidence:** 2

**Review:**

**Summary:**

The paper introduces RecCrysFormer, a hybrid model that leverages transformers and convolutional layers to predict protein structures from X-ray crystallographic data. By processing Patterson maps and partial protein structures, the model predicts electron density maps, which are essential for atomic-level protein data modeling. The paper also introduces a novel recycling training strategy that iteratively refines predictions by incorporating outputs from previous runs and crystallographic refinement software. Experimental results on synthetic datasets demonstrate its accuracy and robustness, marking progress towards integrating machine learning with experimental crystallography for protein structure determination.

**Pros:**
- The paper is well-written and well-motivated overall.
- Leveraging ML and large-scale foundation models to enhance structural biology is a promising research direction.
- The results of the proposed model appear promising (though the reviewer does not have sufficient domain knowledge to fully assess them).

**Cons:**
- Overall, the paper is solid and falls within the computational biology and AI for Science domain. However, I must admit a lack of sufficient domain knowledge to assess some major aspects of the paper. A potential concern is that RecCrysFormer has not been compared with other state-of-the-art models in the field. Is this because the proposed architecture is the first work for protein structure prediction from X-ray crystallographic data? If not, it is essential to include baseline comparisons.
- It would also be beneficial to conduct a scaling law study on the proposed model to understand its scalability with currently available data.

---

### Meta-Review · Area_Chair_BkRd · 2025-02-05

**Recommendation:** Accept (Poster)
**Confidence:** 4

**Metareview:**

The paper studies a hybrid model with transformers and convolutional layers to predict protein structures from X-ray crystallographic data. The paper also introduces a recycling training strategy that could iteratively refine predictions by incorporating outputs from previous runs and crystallographic refinement software. Experimental results on synthetic datasets demonstrate good performance. Reviewers acknowledge the significance of this research.

---

### Decision · Program_Chairs · 2025-02-11

Accept (Poster)